# Application of High-Speed Gallium Nitride Devices in Mass Spectrometry Sweeping Mode

Le Han [1,2,3,4] and Yongping Li [1,2,3,4,*]

1 National Space Science Center, Chinese Academy of Sciences, Beijing 100190, China
2 University of Chinese Academy of Sciences, Beijing 100049, China
3 Beijing Key Laboratory of Space Environment Exploration, Beijing 100190, China
4 Key Laboratory of Science and Technology on Environmental Space Situation Awareness, Chinese Academy of Sciences, Beijing 100190, China
* Correspondence: lyp@nssc.ac.cn

**Abstract:** Quadrupole mass spectrometers are widely used, and voltage scanning is their traditional working mode. By fixing the scanning voltage frequency and changing the value of the RF voltage, ions with different mass numbers can reach the detector in sequence, achieving ion selection. When analyzing high-mass molecules, several kilovolts of scanning voltage are required, which is not conducive to the miniaturization and safety of the instrument. By selecting the frequency of the scanning RF power supply and fixing the value of the RF power supply voltage, ion selection can be achieved by changing the frequency of the RF power supply, enabling miniaturized mass spectrometry analysis of high-mass molecules. In this paper, a high-speed gallium nitride driver circuit for frequency scanning mass spectrometry analysis is designed. The NCP51820 high-speed gate driver and INN650D140A gallium nitride MOS tube are selected to form a full-bridge driver, realizing a quadrupole rectangular wave RF power supply. The system has a maximum withstand voltage of 650 V and a frequency range of 400 K–4 MHz, allowing for scanning measurements of mass numbers ranging from 3 to 606 amu.

**Keywords:** gallium nitride; quadrupole; fixed-voltage variable frequency; mass spectrometry analysis





## 1. Introduction

A mass spectrometer is an analytical instrument used for analyzing and detecting the composition of substances. It can separate and detect ions based on their mass-to-charge ratio, enabling analysis of the composition [1]. It is widely used in various fields such as physics, chemistry, geology, and astronomy [2–4].

A quadrupole mass spectrometer is a commonly used type of mass spectrometer today. Compared to other mass spectrometers, it is compact and has a simpler manufacturing process. It requires a lower vacuum level and is suitable for a wider range of scenarios. Whether ions can reach the downstream detection region through the quadrupole field is mainly determined by the performance of the RF power supply [5]. The DC voltage, AC voltage, and frequency of the RF power supply are the main indicators for evaluating the performance of the RF power supply.

The traditional method of fixed-voltage scanning frequency requires a high circuit switching speed to achieve good performance. However, the switching frequency of conventional MOS transistors is not high enough. Therefore, most existing quadrupole mass spectrometers are based on the working mode of fixed frequency scanning amplitude [6]. After selecting a fixed frequency, a scanning voltage of several kilovolts is used for mass spectrometry measurements, which brings complexity and difficulties to circuit design. With the development of gallium nitride manufacturing and application technology in recent years, the use of gallium nitride power transistors has become more mature, with ultra-high switching speeds and voltage withstand capability. Therefore, in this paper,

based on the working principle of constant pressure scanning frequency in mass spectrometry, a new high-frequency scanning module is designed by combining high-frequency gallium nitride devices, providing a new implementation method for the RF power supply of mass spectrometry analysis.

## 2. Quadrupole Mass Spectrometer

A schematic diagram of a quadrupole is shown in Figure 1. Its principle is based on the fact that when ions move in the quadrupole, their trajectories differ due to the specific combination of the AC and DC electric fields, depending on their mass-to-charge ratio. This allows for the analysis of the composition of substances [5]. As it operates solely on electric fields and does not involve magnetic fields, the quadrupole has the advantages of simplicity, compact size, and fast scanning capabilities [7].

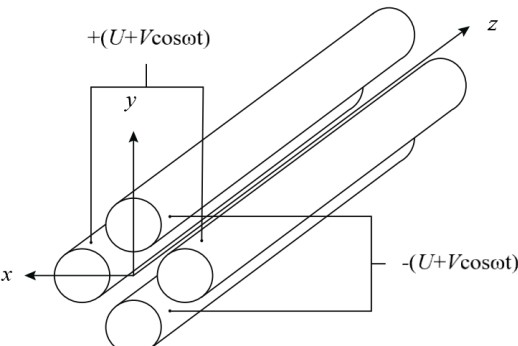

**Figure 1.** Structure diagram of quadrupole mass spectrometer.

A quadrupole structure consists of four parallel rods with equal spacing and length. The rods are supplied with both DC and AC voltages, and the opposite parallel rods are connected in series. High-frequency high-voltage signals with the same amplitude and a phase difference of 180° are superimposed on the electrodes in the XY direction. By controlling the DC voltage, frequency, and AC voltage of the high-frequency signal, ions of specific masses can oscillate stably within the quadrupole field [8–10].

The motion of ions in the quadrupole field follows Newton's equation of motion, $F = ma$. By setting $\xi = \omega t/2$, the ion's motion can be normalized into the Mathieu equation [11].

$$\frac{d^2u}{d\xi^2} + (a_u - 2q_u cos2\xi)u = 0 \tag{1}$$

The mathematical form of Equation (1) represents parametric oscillation, where the ion's motion frequency and mode in the quadrupole can be altered by changing the parameters $a_u$ and $q_u$.

$$a_u = a_x = -a_y = \frac{8zeU}{m\omega^2r^2} \tag{2}$$

$$q_u = q_x = -q_y = \frac{4zeV}{m\omega^2r^2} \tag{3}$$

where $a_u$ is a parameter in the Mathieu equation related to the DC electric field, $q_u$ is a parameter in the Mathieu equation related to the AC electric field, $z$ is the charge number, $U$ is the DC voltage, $V$ is the AC amplitude, and m is the ion mass.

By changing the values of $a_u$ and $q_u$, the trajectory of ions in the quadrupole electric field can be determined, and the stable and unstable boundaries of ions in space can be represented in the $a_u$-$q_u$ coordinate system, as shown in the figure. The stable region represents the electric field conditions that keep the ions stably present in the analyzer, while the unstable region represents the electric field conditions that eject the ions from the analyzer. According to the Mathieu equation, when ions are in the stable region, their

motion trajectory tends to be harmonic. When ions are in the unstable region, the motion amplitude of ions with a large mass-to-charge ratio decays rapidly in an exponential form, while the motion amplitude of ions with a small mass-to-charge ratio increases rapidly in an exponential form. In either case, the ions will eventually leave the equilibrium field, collide with the quadrupole rod, or be ejected from the electric field, making it impossible for the ion detector to detect them.

An enlarged image of the first stable region is shown in Figure 2, where the green area formed by the overlap of two stable regions represents the first stable region. In actual instrument operation, only the first quadrant is used, and only the ion motion within this region is stable periodic motion, rather than divergent motion.

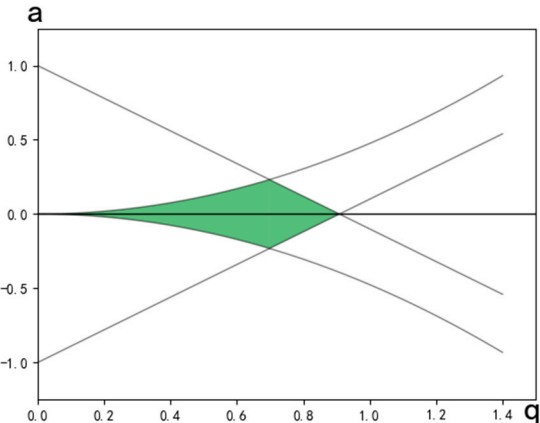

**Figure 2.** An enlarged image of the first stable region.

The values of the Mathieu equation parameters and the first stability region will change depending on the type of power source [12,13]. Here, we will discuss it using sine waves and square waves.

Under a sine wave RF power source, the values of the Mathieu equation parameters are $0 < a_u < 0.2369$ and $0 < q_u < 0.908$. The vertex $(q_u, a_u)$ is located at $(0.706, 0.2369)$, as depicted in Figure 3.

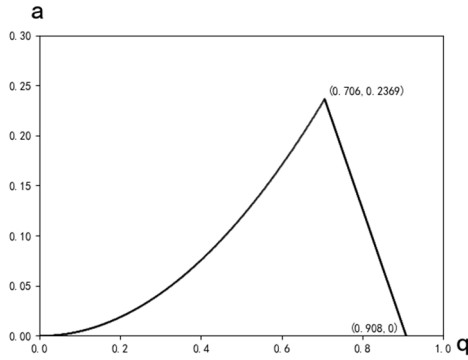

**Figure 3.** Stability diagram of ion motion equation for sine wave power supply.

When driving a quadrupole with a square wave with a duty cycle of 50%, the values of the Mathieu equation parameters are $0 < a_u < 0.2381$ and $0 < q_u < 0.7125$. The vertex $(q_u, a_u)$ is located at $(0.5529, 0.2369)$, as depicted in Figure 4.

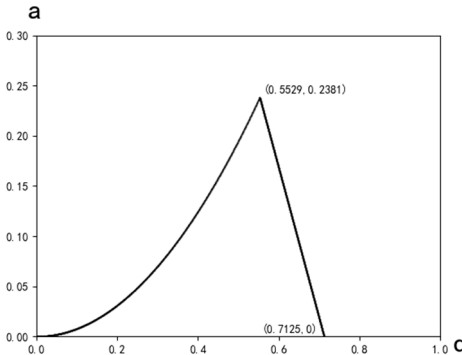

**Figure 4.** Stability diagram of ion motion equation in matrix wave power supply.

### 3. RF Power Supply

The RF power supply of a quadrupole mainly consists of two methods. The first method is to scan the voltage using a fixed frequency of the power supply, while the second method is to scan the frequency using a fixed voltage of the power supply. The method of scanning the voltage with a fixed-frequency power supply is the main working mode used in traditional quadrupole mass spectrometers. It usually uses an oscillating circuit with a quartz crystal or digital frequency synthesis technology to generate the required high-frequency signal. The signal is then amplified and waveform-modulated using a power circuit [14]. After the circuit generates a high-frequency signal, it is input to a transformer for voltage boosting. By linearly changing the input voltage of the transformer, the output voltage of the transformer's secondary stage changes. During this process, both the AC voltage V and the DC voltage U of the RF power supply will change, and the voltage changes are transmitted to the electrodes of the quadrupole, causing changes in the electric field. Ions move in the electric field of the quadrupole, and if they meet the conditions for stable motion, they will sequentially reach the detector. The detector collects ions and generates a voltage, thus achieving ion selection.

Regarding the working principle of the RF power source in fixed-frequency sweep mode, it can be derived as follows by rearranging Equations (2) and (3) to Equation (4):

$$\begin{cases} V = \dfrac{m}{e}\pi^2 f^2\, r_0{}^2 q_u \\[2mm] U = \dfrac{a_u V}{2q_u} \end{cases} \tag{4}$$

By substituting the parameters from Table 1 into Equation (4), we can obtain a *U-V* stability diagram for ions with different mass-to-charge ratios (*m/e*):

**Table 1.** Parameters under scanning mode.

| Parameters | Values | Units |
|:---:|:---:|:---:|
| Frequency of scanning $f$ | 2 | MHz |
| Diameter $r_0$ | 2.7 | mm |
| Ion mass number | $n*1.6605*10^{-27}$ | KG |
| $e$ | $1.6022*10^{-19}$ | C |
| $a_u$ | 0.2369 | |
| $q_u$ | 0.706 | |

As shown in Figure 5, in this study, we only consider the values of the vertex in the first stability region, i.e., the vertex ($q_u$, $a_u$). This point ensures the limit parameters for the stable passage of ions with a specific mass-to-charge ratio through the quadrupole field. By calculating the values at this vertex, we can determine the *U-V* amplitude corresponding to each mass-to-charge ratio. For a set of mass-to-charge ratios [5, 30, 70, 150, 220, 500,

1000], the larger the mass-to-charge ratio, the higher the required scanning voltage. For example, when the scanning mass-to-charge ratio is 1000, an alternating voltage of 2102 V is needed. However, achieving such a high AC voltage of 2102 V poses circuit complexity and difficulties.

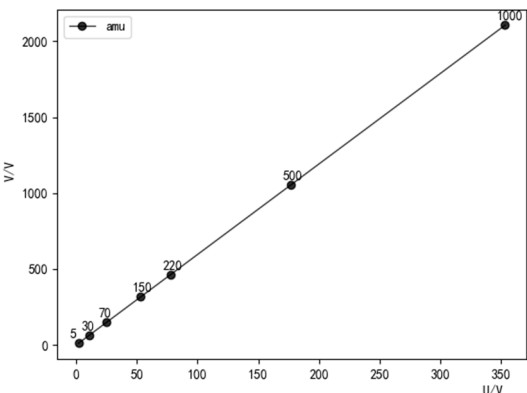

**Figure 5.** *U-V* stability diagram.

On the other hand, for frequency scanning of the quadrupole RF power source, the advantage lies in not requiring extremely high voltages for scanning ions with ultra-high mass-to-charge ratios. Instead, a fixed voltage can be used to change the frequency and achieve the same scanning effect. To establish the equation relating the mass-to-charge ratio and frequency for frequency scanning, we can rearrange Equations (2) and (3) to obtain Equation (5).

$$\begin{cases} U = \dfrac{V}{2} * \dfrac{a_x}{q_x} \\ f = \sqrt{\dfrac{eV}{m}} * \dfrac{1}{\pi r_0} * \dfrac{1}{\sqrt{q_x}} \end{cases} \tag{5}$$

By selecting a fixed U-V voltage and varying the radio frequency (RF) f, we can achieve ion selection using Equation (5). By substituting the parameters from Table 2 into the equation, we can obtain a *U-f* stability diagram for ions with different mass-to-charge ratios (*m/e*).

**Table 2.** Parameters under frequency sweep mode.

| Parameters | Values | Units |
|---|---|---|
| Voltage of scanning $f$ | 40 | V |
| Diameter $r_0$ | 2.7 | mm |
| Ion mass number | $n*1.6605*10^{-27}$ | KG |
| $e$ | $1.6022*10^{-19}$ | C |
| $a_u$ | 0.2369 | |
| $q_u$ | 0.706 | |

As shown in Figure 6, when selecting a scanning voltage of 40 V and the same mass-to-charge ratios [5, 30, 70, 150, 220, 500, 1000], scanning for ions with larger mass-to-charge ratios can be achieved with only 8.61 V using the square wave RF power source scanning method. For ions with a mass-to-charge ratio of around 30 amu, the required frequency is 1.8 MHz.

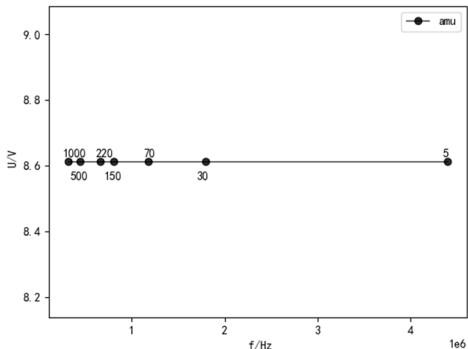

**Figure 6.** *U-f* stability diagram.

For the same quality number scanning width, if the method of scanning the voltage using a fixed frequency is chosen, it would require generating kilovolts of voltage. The higher the voltage, the higher the design requirements for the circuit and the more complex the circuit becomes. On the other hand, if the scanning frequency is achieved by using a fixed RF power supply voltage, only a voltage within 100 V is needed to achieve the same effect. Compared to the high voltage of kilovolts, circuits with low voltages are easier to implement.

## 4. RF Power Supply Design

### 4.1. Overall Block Diagram

This paper mainly focuses on the driver and GaN chip section, specifically the driver circuit part. An external variable DC power supply (providing AC voltage V) is used, and an FPGA is chosen as the controller. The controller outputs two complementary signals with a dead zone to control the driver. The driver controls the on/off state of the GaN in the later stage, resulting in a high-frequency square wave voltage with the same amplitude and a phase difference of 180°. This achieves a square wave AC power supply for the quadrupole RF power source, as depicted in Figure 7.

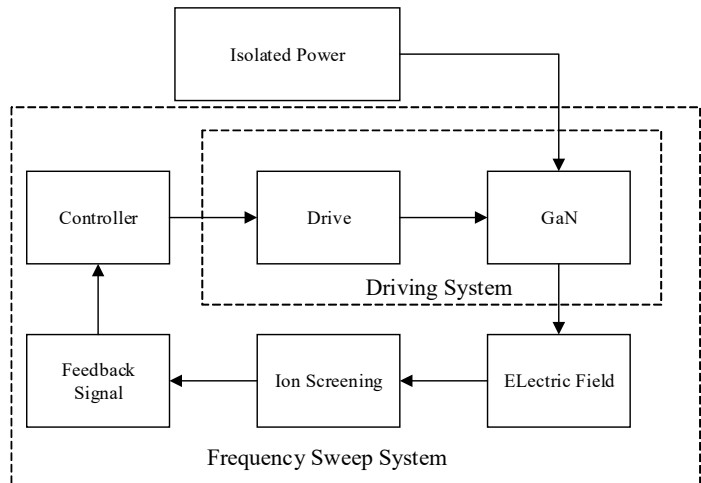

**Figure 7.** Overall block diagram of scheme design.

### 4.2. GaN Device

With the rapid development of the semiconductor industry, the device options for fast-switching circuits have expanded beyond transistors, Si MOS, and IGBT to include GaN MOS. Compared to traditional Si MOS, GaN MOS devices have advantages such as high-frequency characteristics, high reliability, and low power consumption. The key advantage is that the switching frequency of GaN MOS devices can reach tens of MHz,

compared to the MHz range limit of Si MOS. This characteristic meets the high voltage and high-frequency requirements of the square wave power supply in this system. The pinout diagrams of traditional MOSFET and GaN are shown in Figure 8.

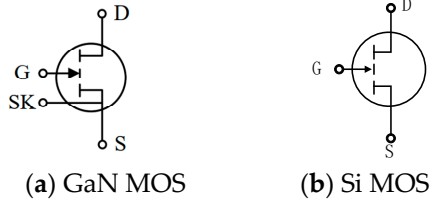

(**a**) GaN MOS          (**b**) Si MOS

**Figure 8.** Schematic diagram of MOS tube pins.

Differences between Si MOS and GaN:

- Drive voltage: The drive voltage for Si MOSFET is 12 V, while for GaN it is +6 V.
- Kelvin source pin: GaN MOS has an additional SK pin compared to Si MOS, which avoids the sharing of source lines between the drive circuit and power circuit, achieving decoupling of the two circuits.

The GaN device chosen for this design is an INN650D140A chip from Infineon Technologies. It is a 650 V silicon-based GaN enhancement mode power transistor packaged in DFN 8 mm × 8 mm. Its parameters are as follows Table 3:

**Table 3.** INN650D140A parameter table.

| Parameters | Values | Units |
|---|---|---|
| VDS, max | 650 | V |
| RDS(on), max @VGS = 6 V | 140 | mΩ |
| QG, typ @ VDS = 400 V | 3.5 | nC |
| ID, pulse | 32 | A |
| QOSS @ VDS = 400 V | 33 | nC |
| Qrr @ VDS = 400 V | 0 | nC |
| Turn-on delay time | 3 | nS |
| Turn-off delay time | 4 | nS |
| Rise time | 5 | nS |
| Fall time | 4 | nS |

The INN650D140A has a voltage rating of 650 V and a maximum current of 32 A. The delay time for turn-on and the rise/fall time for this GaN MOS device are in the nanosecond range. Compared to Si MOS, it offers significant improvements in voltage rating and switching frequency. These characteristics make it suitable for meeting the requirements of this design.

### 4.3. GaN Gate Driver

We selected an NCP51820 high-speed gate driver chip as the driver for the four-pole square wave RF power supply. This IC possesses advanced level shifting and level shifting capabilities, along with an exceptionally fast propagation delay. With a simple circuit, the gate of the MOSFET in the upper bridge arm can support a common mode voltage range of 3.5 V to +650 V, significantly enhancing the voltage withstand capability of the power supply. On the other hand, the lower bridge arm can provide a common mode voltage range of −3.5 V to +3.5 V, enabling negative voltage capability and enabling faster turn-off of the MOSFET, achieving a response rate of 200 V/ns. The pinout diagram of the NCP51820 chip is shown in Figure 9.

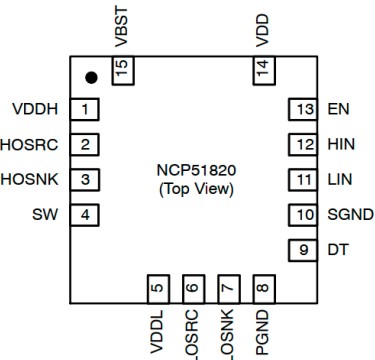

**Figure 9.** NCP51820 pin diagram.

Furthermore, this chip also features an independent undervoltage lockout function, which provides protection to the circuit when the voltage falls below the threshold. It also offers protection against circuit overheating. Additionally, the dead time of the MOSFET can be set through resistor programming.

As shown in Figure 10, the NCP5120 chip differs from traditional gate driver chips in two aspects. Firstly, it uses two pins to jointly control the gate of the MOSFET, with one pin controlling the gate drive pull current and the other controlling the gate drive push current. This helps to accelerate the turn-on and turn-off time of the MOSFET. Secondly, it incorporates a Kelvin source pin, which is used to isolate the gate drive return current from the higher current and voltage levels that may occur at the high-side power switch node or the low-side power supply ground.

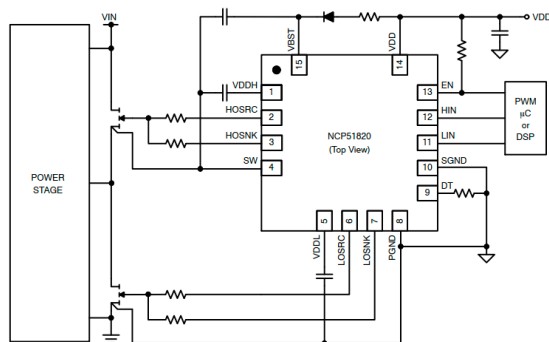

**Figure 10.** NCP51820 drive diagram.

The chip has a programmable dead time feature, which can be set using a resistor connected to pin 9 of the chip. When the resistor is less than 25 k, the dead time is provided by the input signal. When the dead time is greater than 0 ns, the dead time signal is transmitted. When the dead time is less than or equal to 0 ns, the chip provides protection to prevent short circuits in the output stage. When the resistor is between 25 k and 200 k, the dead time can be fixed at 25 ns to 200 ns, determined by the resistor. When a 249 k resistor is used, the dead time is fixed at 200 ns and cannot be changed. When pin 9 of the chip is connected to VDD, both the upper and lower bridge arms are allowed to conduct simultaneously. This is dangerous for half-bridge applications and is mainly suitable for scenarios where both bridge arms need to be conducted or turned off simultaneously. In this design, the exploration of using the input signal to control the dead time is chosen, which allows for faster switching frequency.

When designing the circuit, it is important to distinguish between the signal ground (SGND) and the power ground (PGND). These two grounds are isolated from each other within the chip. The PGND is mainly used for signal control in the circuit, controlling the gate driver. On the other hand, the SGND is the ground for the power current, where the circuit's current primarily flows through. In PCB design, the PGND should be placed as

close as possible to the NCP51820 to reduce the impedance of the control signal. SGND traces should be wider to prevent overheating caused by excessive current in the output stage.

To build the peripheral circuit using the circuit shown in the diagram, use appropriately sized resistors and capacitors. Pull the EN pin high to enable the chip. At this point, use a signal generator or controller to input two complementary PWM signals with dead time to the HIN and LIN pins. This will enable and disable the upper and lower arms of the gallium nitride MOSFET in the output stage, allowing for power control.

### 4.4. Full Bridge Drive Circuit

After setting up a single NCP51820 half-bridge system, it is necessary to use two sets of NCP51820 and GaN chips to form a full-bridge power circuit. One should connect the two outputs of the full-bridge driver to the two poles of the four-pole to achieve a complementary square wave output, as shown in Figure 11.

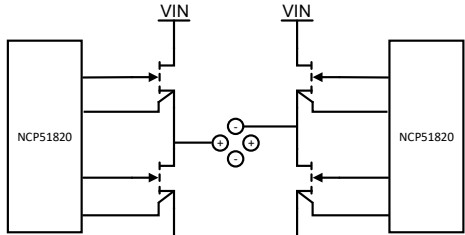

**Figure 11.** Schematic diagram of quadrupole RF power supply.

The circuit is implemented using two sets of an NCP51820 in conjunction with a GaN MOS to form a bridge circuit. At different time points, the upper and lower bridge arms of the two paths are controlled to achieve a positive and negative power supply effect, which is then applied to the two parallel rods of the quadrupole to form a rectangular wave RF power supply V.

## 5. Theoretical Parameter Calculation

Before conducting the physical construction and testing, theoretical parameter calculations are performed. Based on Equations (2) and (3), the relationship between the mass number and U, V, and f can be obtained as Equation (6).

$$\begin{cases} \dfrac{m}{e} = \dfrac{V}{\pi^2 r_0 f^2 q_u} \\[2mm] U = \dfrac{V}{2} * \dfrac{a_u}{q_u} \end{cases} \tag{6}$$

For this design, assuming a rectangular wave $V = 40$ V, $r_0 = 2.7$ mm, $e = 1.6*10^{-19}$, $f = 400$ KHz, $a_u = 0.2369$, $q_u = 0.5529$, and $m = n*1.6605*10^{-27}$, we can calculate the value of $n$.

$$n = 606.41 \tag{7}$$

By calculation, we find that $n = 606.41$. According to this formula, different frequencies $f$ (400 K–4 MHz) are substituted into the equation to obtain different mass numbers m. Based on this, a graph is plotted as shown in Figure 12.

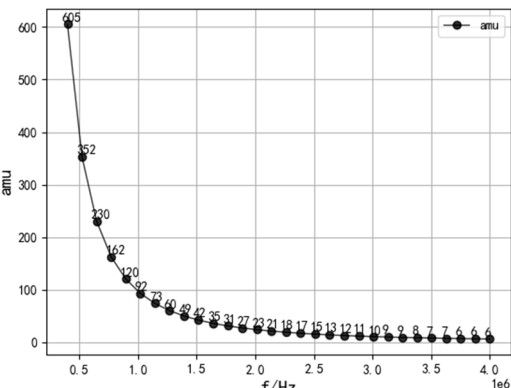

**Figure 12.** Relationship between frequency and measurable limit mass number *m*.

From Figure 12, it can be observed that when the frequency reaches 4 MHz, the measurable limit of the mass number is six. When the frequency is 400 K, the measurable mass number is 606 amu.

Assuming V = 20 V, with other conditions remaining unchanged, the relationship between the mass number and frequency is shown in the following graph.

From Figure 13, it can be observed that when the frequency reaches 4 MHz, the measurable limit of the mass number is three. When the frequency is 400 K, the measurable mass number is 302 amu. Therefore, the theoretical scanning width of the mass number for this system is 3–302 amu. Furthermore, several voltage and frequency values corresponding to specific mass numbers are listed, as depicted in Table 4

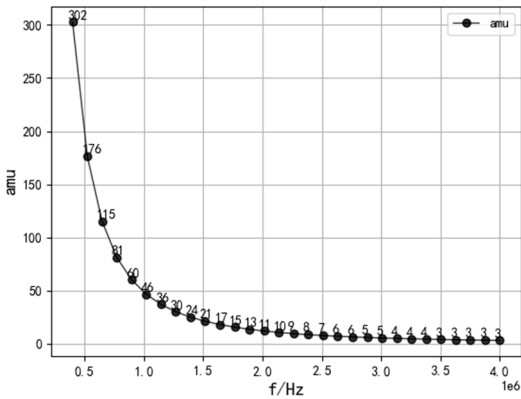

**Figure 13.** Relationship between frequency and measurable limit mass number *m*.

**Table 4.** Mass number of different voltages and frequencies.

| AC Voltage | Frequency/Hz | Mass Number/amu |
|---|---|---|
| 40 | 400 K | 606 |
| 40 | 1000 K | 97 |
| 40 | 4000 K | 6 |
| 20 | 400 K | 302 |
| 20 | 1000 K | 48 |
| 20 | 4000 K | 3 |

## 6. Practical Testing

### 6.1. Printed-Circuit Board

After PCB fabrication and soldering, the soldering process is performed to obtain Figure 14. The signal input and the 12 V power supply for the chip are placed on the right side of the board for easy wire connection. Various capacitors and resistors used by the NCP51820 are placed closely to the chip to reduce the impact of a long circuit on the signal.

The GaN chip, driver power input, and full-bridge driver output are placed on the left side of the board. When drawing this part, the width of the wires needs to be increased to prevent excessive current and heating due to a narrow wire width. A capacitor is placed at both ends of the driver input power supply to filter and stabilize the power supply.

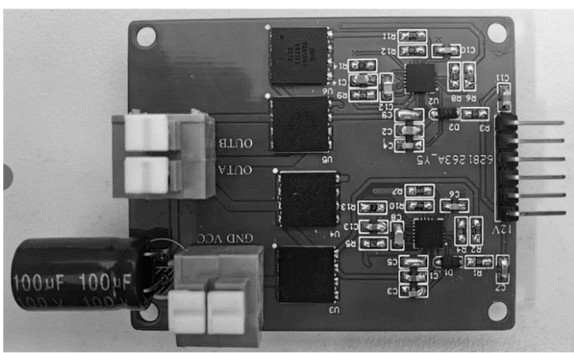

**Figure 14.** PCB image.

*6.2. Complementary PWM Input Waveform*

The controller outputs a 4 MHz complementary square wave. The oscilloscope clamps the signal at the input terminals, resulting in Figure 15. Before and after each cycle, both signals are pulled low for a certain period of time, which is called the dead zone. In order to prevent simultaneous conduction of the upper and lower bridge arms in the bridge circuit, it is necessary to pull both sides low for a certain period of time when flipping the state of the upper and lower bridge arms in the control to avoid short circuits and damage to the circuit. The actual measured frequency is 4.000 MHz with an amplitude of 3.3 V, meeting the design requirements.

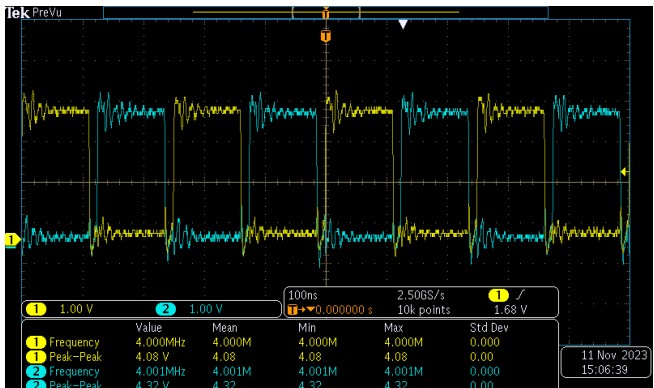

**Figure 15.** A 4 MHz dual complementary.

*6.3. Oscilloscope Graph*

The oscilloscope measures the drive output using two probes grounded together to clamp the output of the bridge circuit, resulting in the following waveform diagram:

Figure 16 shows the test waveform of an AC voltage of 10 V at a frequency of 4 MHz. The measured frequency is 4.018 MHz, and the peak-to-peak value is 20.8 V.

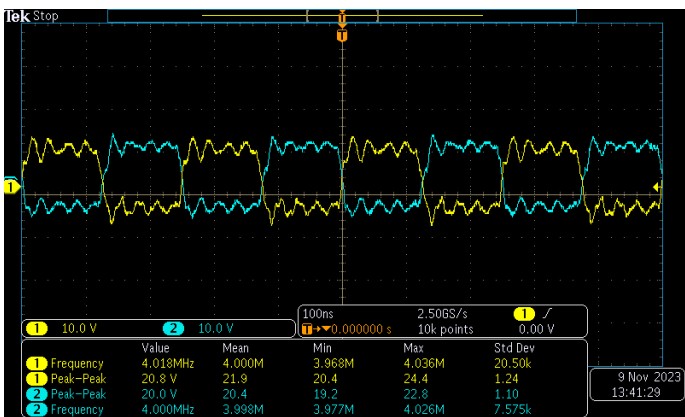

**Figure 16.** A 10 V 4 MHz output.

Figure 17 shows the test waveform of an AC voltage of 20 V at a frequency of 4 MHz. The measured frequency is 4.016 MHz, and the peak-to-peak value is 40.4 V.

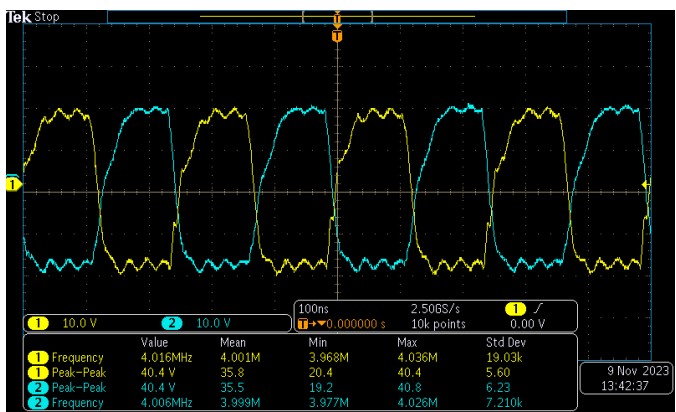

**Figure 17.** A 20 V 4 MHz output.

Figure 18 shows the test waveform at a frequency of 4 MHz with an AC voltage of 40 V. The measured frequency is 4.058 MHz, and the peak-to-peak value is 79.2 V.

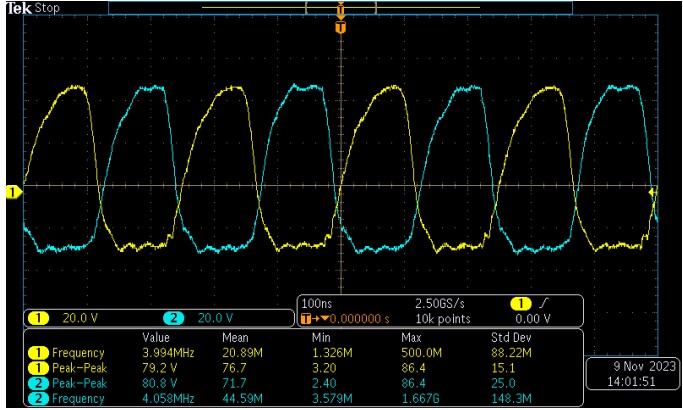

**Figure 18.** A 40 V 4 MHz output.

The measurement method mentioned above tests the peak-to-peak value of the driving circuit, and the corresponding amplitude of the RF power supply AC voltage needs to be divided by two. For example, an 80 V peak-to-peak value corresponds to a 40 V amplitude of the RF power supply AC voltage. Actual tests were conducted using RF power supply

AC voltages of 10 V, 20 V, and 40 V at 4 MHz, and all showed good rectangular wave power supply outputs, meeting the design requirements of the RF power supply.

## 7. Conclusions

The novel gallium nitride-based mass spectrometry scanning circuit designed in this study can achieve high-frequency scanning voltage outputs of 4 MHz 20 V, 4 MHz 40 V, and 4 MHz 80 V. The square wave can achieve the scanning of ion mass numbers ranging from 3 to 305 amu between a 20 V AC voltage and 400 K–4 MHz frequency. Similarly, between a 40 V AC voltage and 400 K–4 MHz frequency, the scanning of ion mass numbers ranging from 6 to 606 amu can be achieved, covering a scanning range of 3–606 amu mass numbers. Since the maximum voltage of the driving circuit is 650V, the limit parameters can be further optimized. The circuit will be tested in the mass spectrometry system to expand the measurement upper and lower limits.

For a mass spectrometer, resolution is also crucial. However, in this design, a square wave RF power supply is used instead of a sine wave. It is currently unclear whether the resolution calculation formula for sine wave RF power supplies is applicable to square wave RF power supplies, so this aspect is not mentioned in this paper. However, for a sine wave RF power supply, the resolution of a quadrupole mass spectrometer mainly depends on the slope of the mass scanning line, which is primarily determined by the values of $a_u$ and $q_u$. In this design, the values of $a_u$ and $q_u$ can be adjusted by changing the voltage of the superimposed DC power supply, allowing for an adjustable resolution. Compared to RF power supplies with a fixed-frequency scanning amplitude, this power supply can achieve finer scanning and higher resolution by selecting different external driving voltages and frequencies based on the ions to be scanned.

The proposed driving circuit has a simple structure and reliable performance and is capable of achieving high voltages and frequencies. The generated square wave power supply can meet the requirements of quadrupole scanning mass spectrometry analysis.

**Author Contributions:** Formal analysis, L.H. and Y.L.; investigation, L.H. and Y.L.; methodology, Y.L.; software, L.H.; validation, L.H. and Y.L.; writing—original draft, L.H.; writing—review and editing, L.H. All authors have read and agreed to the published version of the manuscript.

**Funding:** This research was funded by the Chinese Meridian Project, grant number Y91GJC15ES and grant number Y91GJC15DS. This research was funded by the Civil Aerospace Pre-Research Project, grant number E266000102. The APC was funded by the National Space Science Center, Chinese Academy of Sciences.

**Data Availability Statement:** Data are contained within the article.

**Conflicts of Interest:** The authors declare no conflict of interest.

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
