# Peer review of "Application of High-Speed Gallium Nitride Devices in Mass Spectrometry Sweeping Mode"

_electronics, doi:10.3390/electronics12244966_

Round 1
Reviewer 1 Report
Comments and Suggestions for Authors
Summary:
The authors designed a new high-frequency scanning module with the recently developed gallium nitride devices. By tuning the voltage and frequency, scanning of ion mass numbers between 3-606 amu can be achieved.
Impact:
The authors applied novel GaN devices to develop a new RF power supply in mass spectroscopy analysis. This paper could be interesting to the readers of Electronics, especially for the researchers working on spectroscopy, small molecules, atomic clusters, and astronomy. However, more introduction and discussion need to be included before this paper can be published.
Issues:
1. The introduction of this manuscript needs to be improved. As the authors are talking about the scanning module for quadrupole mass spectrometer, the introduction should talk about the following: A) The development and application of the quadrupole mass spectrometer, as well as its pros and cons compared to other mass spectrometers, like time of flight. B) The development of RF power supply and the challenges limiting its performance. C) The recent development of GaN devices and how it would make an impact.
2. Similar to point 1, after presenting all the results, the authors should discuss more about the novelty and improvement of their design in the conclusion part. A comparison with the commonly used RF power supplies nowadays would be helpful.
3. In addition to the range of mass numbers, resolution is also very important in mass spectroscopic studies. What resolution can this design reach in different mass ranges? In what range can the resolution reach 1 amu? Will it be easy to scan different frequencies and tune the resolution?
Reviewer 2 Report
Comments and Suggestions for Authors
Han and Li present the design of a novel chip based on GaN capable of tackling frequency sweeps for quadrupole spectroscopy. It is unclear, based on what they present, how much of an improvement this design represents with respect to what it is already out in the market. This is a fundamental aspect which the authors should clarify. If my interpretation is correct, and the presented chip offers advantages, then it definitely grants publication in Electronics. Apart from that, I have a few other comments with the authors should address before acceptance.
Overall, the article is poorly organised, in my view. It took me reading it whole to understand that the authors were presenting a novel GaN based MOS design. This should be clearly stated in the abstract, and the introduction should care about explaining the current status of quadrupole spectroscopy, different chips, and the impact of this novel one. Moreover, more emphasis should be put in describing the design, implementation, fabrication, and characteristics of the chip, to allow for easy replication.
Figures are hard to read, with small numbers and difficult interpretation. Specially those regarding the chip characteristics should be more than simply screen captures.
The abstract should not contain U and V that are only to be defined later.
Some subscripts in formulas are simply normal letters in the main text, making interpretation confusing.
Several things are mentioned without definition, for example: stability region, Mathieu equation, or Figure 4 vertex.
Comments on the Quality of English LanguageI am actually unsure whether my difficulties understanding the English in the article stem from the way it is organised, or if there is need for substantial improvement.
Round 2
Reviewer 1 Report
Comments and Suggestions for Authors
The issues have been fixed, this manuscript may be accepted in the present form.
Reviewer 2 Report
Comments and Suggestions for Authors
The authors have made a substantial effort and, consequently, the article is greatly improved, particularly in the clarity of presentation. I still think there are some typos in formulas and how some parameters are used, for example in line 81 au and qu appear like that instead of as au and qu. But these are fine details that can be sorted out during processing of the article. I therefore recommend acceptance.